# LOG REPRESENTATION AS AN INTERFACE FOR LOG PROCESSING APPLICATIONS

## ABSTRACT

Log files from computer systems are ubiquitous and record events, messages, or transactions. Logs are rich containers of data because they can store a sequence of structured textual and numerical data. Many sequential forms of data including natural languages and temporal signals can be represented as logs.

We propose to represent logs at a few levels of abstraction including field level, log level, and log sequence level. The representation for each level can be computed from the previous level. These representations are in vector format and serve as interfaces to downstream applications. We use a version of Transformer Networks (TNs) to encode numerical and textual information that is suitable for log embeddings. We show how a number of log processing applications can be readily solved with our representation.

## 1 INTRODUCTION

A wide range of computer systems record their events as logs. Log-generating systems include telecommunication systems, data centers, software applications, operating systems, sensors, banks, markets, and block-chains ( Barik et al. (2016); Brandt et al. (2020); Busany & Maoz (2016); Cucurull & Puiggalí (2016); Sutton & Samavi (2017)). In these systems, transactions, events, actions, communications, and errors messages are documented as log entries. Log entries are stored as plain text, so they can store any textual or numerical information. A variety of different data types could be viewed as logs, including: natural languages, temporal signals, and even DNA sequences.

All transactions in a Turing machine can be stored as logs. Therefore, logs are theoretically strong enough that they can reproduce the state of any computer system. In some systems log entries are standard and complete. For example, in financial transactions and some database management systems, one can recreate the state of the system by applying a set of rules on the transactions ( Mohan et al. (1992)). In other systems like telecommunication networks log entries are ad hoc, unstructured, and diverse. Therefore, the state of the system cannot be recreated with a set of rules. We will use layered and learnt vector embeddings to represent the state of the system and use them for downstream diagnostic applications including: anomaly detection, classification, causal analysis and search.

**Example:** A snapshot of logs from a telecommunication product installed in a cell tower is shown in Figure 1. The first entry of the log is a trigger for subsequent action of the unit to be restarted. Each log entry will be embedded in a vector space and the sequence of log entrys will have their own vector representation. Timestamp information provides crucial clues about the nature of the event(s).

We summarize our contributions as:

1. We propose levels of abstraction for log representation, in order to standardize and simplify log processing applications.

2. We present a Transformer-based model to embed log sequences in a vector space.

3. We show how log processing applications can be simplified using these log representations. We validate our approach on a real data set obtained from a leading telecommunications vendor. The vocabulary of this data set is twenty times bigger than what is currently available in open source and often used in other research papers[1]

---

[1]After suitable anonymization, we plan to release the data set to the research community.

```
        <HIST> MSABSC4              BCSU-1      SWITCH    2017-03-28  14:44:08.02
*       DISTUR BCSU-1    1A004-00   USAPRO
        (1596) 1023 EXCESSIVE DISTURBANCES IN SUPERVISION
        00 00A0 255d FFFF FFFF

        <HIST> MSABSC4              OMU         SWITCH    2017-03-28  14:44:10.07
        NOTICE BCSU-1    1A004-00   RCXPRO
        (1597) 0691 AUTOMATIC RECOVERY ACTION
        WO-EX SP-RE 0000 0002 7889 00001023 00000000 00000000

        <HIST> MSABSC4              OMU         SWITCH    2017-03-28  14:44:22.09
*       DISTUR BCSU-1    1A004-00   RCXPRO
        (1599) 1001 UNIT RESTARTED
        SP-RE 00 06 00 00 01
```

Figure 1: Sample telecommunication log file including three log entries. Telecommunication logs are complex and diverse because they involve various devices and software.

## 1.1 RELATED WORK

Much of the machine learning oriented work in log analysis has focused primarily on the outlier or anomaly detection problem ( Du et al. (2017); Meng et al. (2019); Du et al. (2019a;b); Yuan et al. (2020); Nedelkoski et al. (2020); Chalapathy & Chawla (2019)).

The prototypical work is DeepLog which, in a fashion analogous to natural language processing, models logs as sequences from a restricted vocabulary following certain patterns and rules ( Du et al. (2017)). An LSTM model $M$ of log executions is inferred from a database of log sequences. To determine if a given element $w_{t+1}$ in a log sequence is normal or anomalous, DeepLog outputs the probability distribution $P_M(\cdot|\mathbf{w_{1:t}})$ where $\mathbf{w_{1:t}} = \langle w_1, w_2, \ldots, w_t \rangle$. If the actual token $w_{t+1}$ is ranked high in $P_M(w_{t+1}|\mathbf{w_{1:t}})$ then it is deemed as a normal event otherwise it is flagged as anomalous. Several variations on the above approach have been proposed. For example, (Yuan et al. (2020)) proposed ADA (Adaptive Deep Log Anomaly Detector) that exploit online deep learning Sahoo et al. (2018) methodology to build on the fly unsupervised adaptive deep log anomaly detector with LSTM. The new models are trained on new log samples.

More recently, Nedelkoski et al. (2020) proposed *Logsy*, a classification-based method to learn effective vector representations of log entries for downstream tasks like anomaly detection. The core idea of the proposed approach is to make use of the easily accessible auxiliary data to supplement the positive (normal) only target training data samples. The auxiliary logs that constitute anomalous data samples can be obtained from the internet. The intuition behind such an approach to anomaly detection is that the auxiliary dataset is sufficiently informative to enhance the representation of the normal and abnormal data, yet diverse enough to regularize against over-fitting and improve generalization.

## 2 SYSTEM LOGS AND NATURAL LANGUAGE MODELS

Log processing models often adopt natural language processing techniques, because logs have several similarities to natural language: (i) Both logs and a natural language consist of a sequence of tokens and (ii) Context matters in both data streams and both models need temporal memory, (iii) Both application scenarios have large datasets and (iv) Annotation is a limiting factor in both settings. This is why log processing literature often reuses natural language approaches.

However there are several differences between system logs and natural languages that need to be brought to the fore: (i) There is temporal information associated with each log entry which can be exploited to give insights about the underlying processes that are generating the logs (ii) Each log entry itself is a composite record of different pieces of information unlike a word in a sentence which is nearly atomic, (iii) Log files often aggregate event logs from multiple threads, processes and devices. Therefore the inference model needs to identify the relevant context among all threads.

### 2.1 SEQ-TO-SEQ MODELS

Transformer Networks (TNs) have become the de-facto choice for sequence-to-sequence modeling( Vaswani et al. (2017)). Based on the concept of self-attention, TNs overcome some of the key limitations of families of Recursive Neural Networks (RNN) including LSTMs and GRUs ( Graves & Schmidhuber (2005)). Transformers Networks can model long range dependencies with constant

$(O(1))$ number of iterations overcoming the RNN limitation where modeling dependencies with distance $n$ takes $O(n)$ iterations. Furthermore since TNs are intrinsically permutation invariant, they can process elements of a log sequence in parallel. This last property is particularly useful in log analysis, perhaps more than NLP, due to the observation made above that log files aggregate event information from different processes. Before we formally describe our extension of Transformer Networks, we introduce the different levels of abstractions suitable for modeling log sequences.

## 3 Levels of Abstraction

In order to standardize and simplify machine learning applications on logs, we propose five levels of abstraction for log representation. Each level of abstraction can be computed from the previous levels.

**1- Log sequence**: A log sequence is a sequence of strings $l_1, l_2, \ldots, l_n$. We refer to each $l_i$ as a *log entry*. Each log entry itself is a sequence of symbols (characters) from an alphabet $\Gamma$.

$$l_i = (c_1, \ldots, c_{m_i}) \text{ where } \forall j, c_j \in \Gamma \tag{1}$$

Log sequences often contain certain parameters including timestamps, IDs and numerical values. In order to access these parameters, we often need to parse logs.

**2- Parsed log**: Whether a log entry has a simple or complex grammar, a log parser will convert it into a set of key-value pairs. Keys represent attributes and values represent the value for each attribute. A parsed log entry can be written as:

$$\{(k_1, v_1), \ldots, (k_{m_i}, v_{m_i}) | k_j \in K, v_j \in V\}. \tag{2}$$

where $K$ is the set of keys (attributes) and $V$ is the set of possible values. Depending on the key, the value could be textual, numerical or categorical. A wide range of log parsers are available and have been evaluated (He et al. (2016); Zhu et al. (2019); He et al. (2017)).

**3- Field Embedding**: A field embedding is a vector representation for a key-value pair. A field embedding function $\rho$ maps a key-value pair into a $D_f$-dimensional vector:

$$\rho : K \times V \longrightarrow \mathbb{R}^D. \tag{3}$$

$D_f$ denotes the dimensionality of field representation. A field embedding function could either be learned or static. They choice of function could depend on key and data type. Since the set of categorical values is often small, categorical values could be stored in a vector quantized format. Similar to many NLP techniques, a look-up table can store vector representations.

**4- Log Embedding**: A log embedding is a vector representation for a single log entry. A log embedding function $\phi$ computes a log representation given its field embeddings:

$$\phi : \mathbb{R}^{D_f} \times \cdots \times \mathbb{R}^{D_f} \longrightarrow \mathbb{R}^{D_l}. \tag{4}$$

$D_l$ denotes the dimensionality of the log embedding. Log embedding could be calculated from field embeddings using DeepSet (Zaheer et al. (2017)) or other means. This log embedding can be used for classification, regression, nearest neighbour search or other algebraic operations.

**5- Sequence Embedding**: A Sequence embedding is a vector representation for a log sequence. This sequence could either be a whole log file, a block of logs, or a sliding window. A sequence embedding $\theta$ function inputs a sequence of log embeddings and outputs a single embedding for the whole sequence:

$$\theta : \mathbb{R}^{D_l} \times \cdots \times \mathbb{R}^{D_l} \longrightarrow \mathbb{R}^{D_s}. \tag{5}$$

$D_l$ denotes the dimensionality of log embedding. The function $\theta$ could be implemented using RNNs or Transformers or other means. We will show that this sequence representation can be used for several downstream operations on log sequences. Some sequence embedding models such as Transformers come with a language model. We show that this language model is also useful in log processing applications.

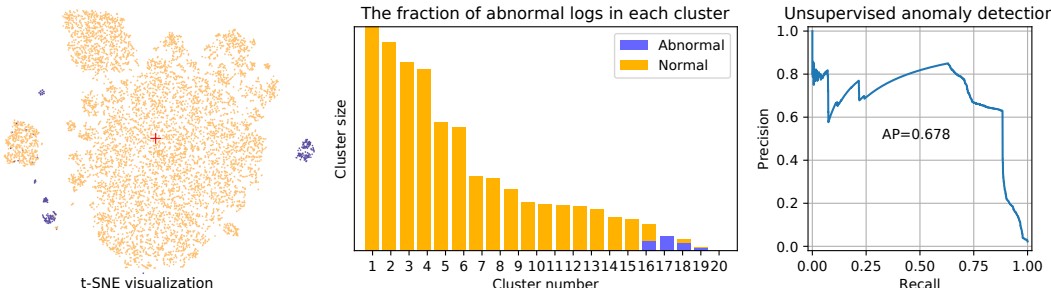

Figure 2: Unsupervised anomaly detection **Left**: t-SNE embedding of our vector representations for log sequences in HDFS dataset. Yellow points mark normal log sequences and blue points mark abnormal log sequences (based on human expert annotation). The red plus sign marks the center of the distribution. Note that abnormal sequences are mostly clustered and are far from the center. **Center**: K-means clustering of our vector representations. Clusters are sorted decreasingly according to their size. Note that abnormal examples are concentrated in the smallest clusters. **Right**: After we sort clusters, we consider cluster number as anomaly score. Here is the P-R curve for predicting anomaies. Note that every step in this pipeline is unsupervised.

## 4 TRANSFORMER NETWORK

We generalize TNs to incorporate the notion of time which is typically available in log data. We use the self-supervised task of masked language model (MLM) to train the network and obtain embeddings. Recall in the MLM task we are given a set of sequences $\{s_i = \langle l_1 l_2 \dots l_{n_i} \rangle | i = 1, \dots, m\}$. For every sequence $s_j$ we randomly select a location $r_j$ in the sequence and then predict $P(l_{r_j} | s_{-r_j})$( Devlin et al. (2018)). While the MLM task by itself may not be particularly useful applications, the embeddings learnt can be fine-tuned to other applications.

Recall transformer networks use the notion of attention to directly estimate the influence of the input context for generating the output token. Attention generalize the notion of the SELECT operation in database query languages. For example give a query $q$ and a table of key-value pairs $\langle k_i, v_i \rangle$, attention is defined $\sum_i \text{sim}(q, k_i) v_i$. The sum is over the input sequence length and sim is the softmaxed similarity between the embedding of the query $q$ and the keys $k_i$.

### 4.1 TIME ENCODINGS

Given a log sequence $\langle (l_1, t_1), \dots, (l_n, t_n) \rangle$ where $l_i$ is the token and $t_i$ is the associated timestamp, we associate a cumulative time $CT(k) = \sum_{j \leq k} (t_j - t_{j-1})$. The Time Encoding (TE) for token position $i$ and dimension $\delta = 1, \dots, d$ is given by

$$TE(i, \delta) = \begin{cases} \sin(\frac{CT(i)}{10000^{2\delta'/d}}) & \text{if } \delta = 2\delta' \\ \cos(\frac{CT(i)}{10000^{2\delta'/d}}) & \text{if } \delta = 2\delta' + 1 \end{cases}$$

Several deep learning architectures are developed to predict masked tokens. Since language models need to have memory and need to handle sequences of unknown length, they have traditionally relied on recurrent models and their variants. TNs have shown several advantages over RNNs and their variants and are now the architecture of choice in many NLP tasks.

We performed an ablation study to evaluate the effect of time encoding (figure 3 left). We trained one log language model that ignores timestamps and one language model that uses timestamps. The language model that is trained with time, performs significantly better than the language model that does not use time. We evaluated this using masked language modeling. We mask a log and try to predict it using nearby words. Our experiments also show that when the timestamp of a masked log is revealed, we can predict the masked word with a significantly smaller error. This confirm that the timestamp is helpful in predicting masked logs.

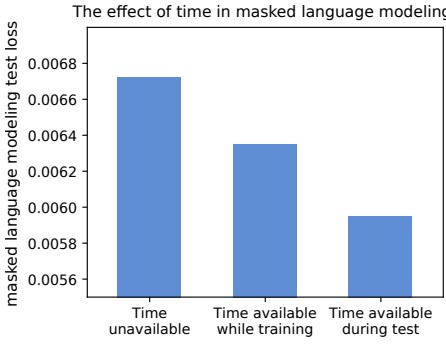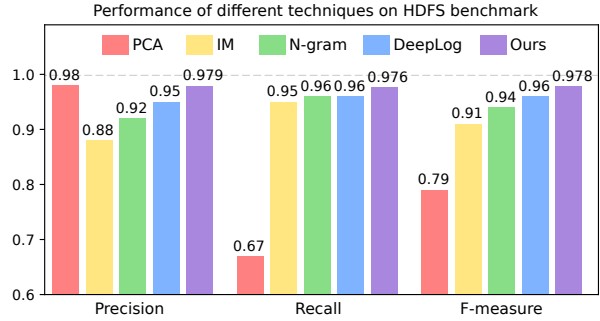

Figure 3: **Left** Masked language modeling loss (cross entropy) during test. Left bar shows test loss using a model that is not trained to use time. Middle bar and right bar show test loss on a model that is trained using timestamp. In the middle bar, timestamp for the masked word is also masked. In the right bar, timestamp for the masked bar is revealed. **Right**: Comparison of different techniques oh HDFS dataset. Our model significantly outperform previous models.

## 5 APPLICATIONS AND EMPIRICAL RESULTS

As soon as a log sequence is embedded into a vector space, many log processing tasks can be expressed as algebraic operations within the vector space. These tasks include: Anomaly detection, Predictive analysis, Causal analysis, Log search, Diagnosis Recommender system and Log generation/synthesis. In this section we quantitatively evaluate the application of the obtained embeddings for these downstream tasks.

### 5.1 DATASETS

We have used two datasets for our experiments: HDFS and a proprietary data from a leading telecommunications vendor.

**HDFS Log Data:** This is an open-source dataset consisting of logs from Hadoop jobs on 200 node Amazon EC2 cluster. There are over 11 million log entrys and around 2.9% are anomalous as labeled by domain experts ( Du et al. (2017)). Tokenization resulted in a vocabulary of size 29.

**Radio Data:** We obtained a real but proprietary data set from a leading telecommunications vendor. The data consists of logs from a product installed in a cell tower for a 4G network. The data set consists of 4783 distinct log files consisting of 1464010 log entries and was collected over a period of two months. This dataset consists of about 860 different log templates and about 180 different software programs.

### 5.2 SOFTWARE AND ARCHITECTURE

We extended the HuggingFace transformer library in pytorch to incorporate time embeddings. We used a simple two layer transformer network, train over ten epochs with batch size of sixteen. In terms of our proposed levels of abstraction (Section 3), we vector quantize log fields (Layer 3) and feed them into the transformer network which constitute Layer 4 and 5 of our proposed abstraction.

### 5.3 ANOMALY DETECTION

Identifying Log Anomalies is crucial for system administration because they could signal errors, system failures, or fraud. Log embedding simplifies the process of anomaly detection. Log embeddings simplify anomaly detection both in unsupervised and supervised settings.

**Unsupervised**: Self-supervised techniques are useful when manual supervision is unavailable. Since our embedding function is learned using a self-supervised model, an application that uses our embedding, automatically benefits from self-supervision (in a transfer learning scheme).

We evaluate our unsupervised anomaly detection technique on HDFS dataset. Given a set of log sequences, We first compute vector representations for each sequence. Then we cluster the examples using k-means. We sort clusters decreasingly according to their size. Since anomalous logs are clustered and rare, they tend to fall into the smallest clusters. We assign an identical anomaly score

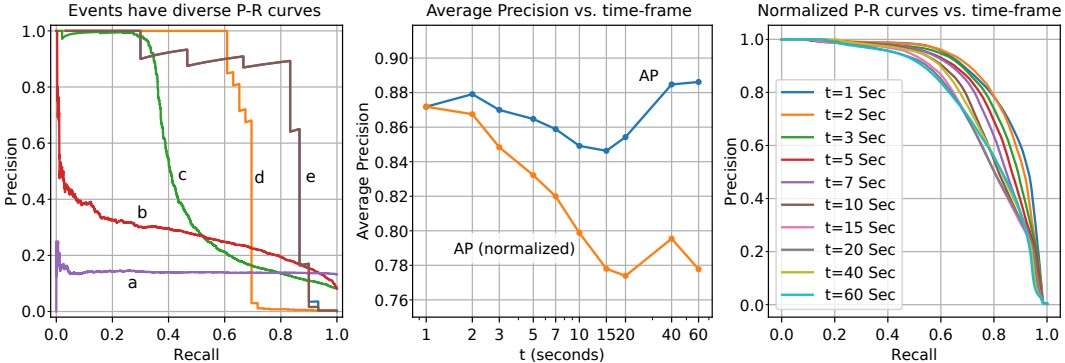

Figure 4: Predicting whether or not an event type will appear within the next $t$ seconds from now. **Left**: Precision-Recall curves for predicting whether an event types will appear within the next $t = 10$ seconds. Note that each of the six different event types has a distinct P-R curve. For example, event types a,b,c are relatively common but harder to predict than d and e. **Middle**: Comparison of P-R curves for all event types across different prediction time-frames $t$. Please note that since longer time-frames are more likely to have an event. Thus, longer time-frames have higher chance baselines. We re-weight test examples to normalize positive-negative ratio and chance baseline. Note that by widening time-scale precision drops. **Right**: Average precision for predicting events at different time-frames $t$. Normalized AP generally decreases by widening time-frame. In Non-normalized AP higher time-frames have higher chance baselines. The increase at $t \geq 40$ seconds is because many processes go through a repeatable shut-down process.

to all points in the same cluster. Larger clusters receive smaller anomaly score and smaller clusters receive larger anomaly score. We evaluate this supervised method in figure 2.

**Supervised**: If anomaly annotations are available, we train a supervised classifier to predict anomalies. Depending on the granularity of annotation (sequence level, time window level, or log entry level), we can extract log embeddings with the same level of granularity. Embeddings could be either context-free or context-dependent depending of the extent of log window. If several modes of anomaly are identified and annotated, classifiers can output scores/probabilities associated with each mode.

We evaluated our supervised anomaly detection model on the HDFS log dataset. Anomaly annotations are given at log sequence level and techniques are compared based on sequence level prediction. In HDFS dataset, sequences are marked with `block_id`. Since anomalous logs are highly clustered, we use SVM with RBF kernel for the classifier. Figure 3 compares our performance with standard baselines on these datasets including PCA, Invariant Mining by Lou et al. (2010), N-grams, and Deeplog by Du et al. (2017).

## 5.4 PREDICTIVE ANALYSIS

Using log embedding we can predict future events including event logs, alarms, transactions, and also their timing and ordering. In figure 4 we demonstrate an example where we predict whether a certain log message is going to appear within the next $t$ seconds.

We first extract embeddings from a sliding window of 64 log entries. Then we assign a target labels to each window. In this example, a target label is a list of log entries that will appear within the next $t$ seconds after the end of the window. We then train a classifier that given the embedding of a log window, tries to predict the list of possible log entries that would appear within the next $t$ seconds. Our classifier predicts a probability score for the appearance of each possible log entry within the next $t$ seconds. A log entry could mark some error, or the beginning or end of a process. Our experiments show that we can reliably predict log entries that will appear within the next few seconds.

This predictive scheme is generic. We can use a similar scheme to predict any event. For example, we can predict whether a system request will fail, how long a transaction will take to complete, what we expect an API to return, or whether a user is going to complete a purchase.

Figure 5: For better readability we have visualized each log template with a letter. **Left**: Given a query log (highlighted blue), we use a language model to predict its probability twice: Once including and once excluding $l_j$. in both cases, log entries prior to $l_j$ are included. **Right**: Visualization of a few examples. Query log is highlighted blue. Previous logs are highlighted green according to their causal score $C(l_j, l_i)$. This visualization helps identify logs that help predict the query log the most.

## 5.5 CAUSAL ANALYSIS

Causal analysis helps track the cause of failures and events. True causality requires controlled experimentation which is often impractical for logs. We use a proxy formulation for causality based on Granger causality (Granger (1969)). We measures the ability to predict a future log using a previous log. Given a log entry $l_j$ at position $j$ and a log entry $l_i$ at position $i$ where $j < i$, we measure how well the knowledge of $l_j$ helps us predict $l_i$. We define $C(l_j, l_i)$ as:

$$C(l_j, l_i) = \log \left( \frac{P(l_i | l_j, l_{j-1}, l_{j-2}, \ldots)}{P(l_i | l_{j-1}, l_{j-2}, \ldots)} \right) \tag{6}$$

Given a log entry $l_i$, we can measure the effect of each of the previous log entries $l_j, j < i$ on our prediction of $P(l_i)$. We use a three-layer deep neural network to predict the probability $P(l_i)$. We found this model to be more accurate than our standard language model, because transformer language models are not explicitly trained to predict several locations in the future.

We did not quantitatively evaluate this causal analysis because no ground-truth is available. However, we illustrate results in figure 5. Depending on use case, variations of this causal analysis can be used. For example, the causal effect of one log field on another log field can be calculated. Or when two parallel log files are generated, the effect of one on another can be calculated.

## 5.6 LOG SEARCH

Some investigative/diagnostic applications involve log search. For example, a debugger could observe an error pattern and needs to find similar patterns in history; An investigator may need to find whether a certain user has followed a certain user journey; Or a financial analyst may need to identify the frequency of a certain pattern.

Nearest neighbor search in the log embedding space can be used for search. Given a log history, we extract log embedding from a sliding window of logs. Then we store the vector representations of each window in a data structure. Given a window of log entries as query, we first compute the log embedding and then search for its nearest neighbors using the data structure.

The choice of data structure affects indexing/retrieval time complexity. One could use a plain array, KD-Tree, LSH-based kNN search or an inverted index. A plain array can handle thousands of log entries, KD-Trees can handle millions of log entries, and inverted index can handle billions of log entries.

Evaluating of log search performance is not trivial because there is no labeled dataset. However, the closest evaluation criteria is based on edit distance (Levenshtein distance). Edit distance is not ideal because it cannot handle time and continuous parameters. Furthermore, each practical application has its unique criteria that is not necessarily identical to edit distance.

Given all these limitations, we chose edit distance because it is concrete and has applications sequence search in NLP and Bioinformatics. The best algorithms for Exact search based on edit distance requires $O(L^2 n)$ time complexity where $L$ is the length of log sequences and $n$ is the number of log windows in the database.

Computing euclidean distance to all examples in log embedding space has a complexity of $O(dn)$, where $d$ is the dimensionality of vector embedding. This complexity is more favorable than that of

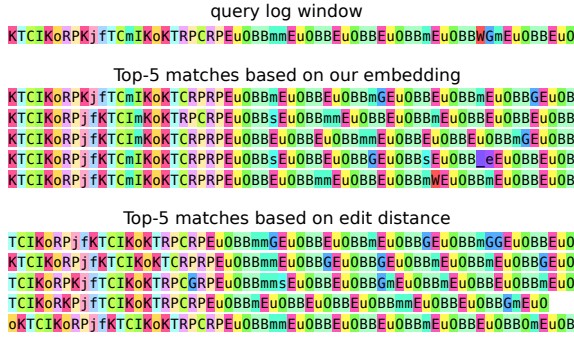
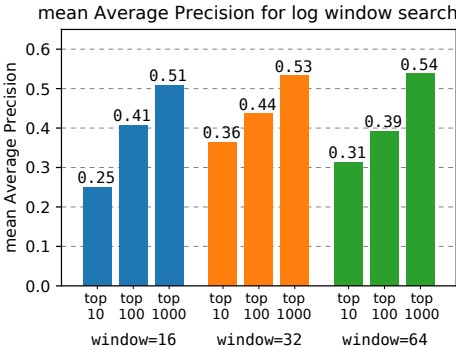

Figure 6: **Left**: A sample query window, top-5 matches based on our embedding and top-5 matches based on edit distance. We have replaced each log template with a unique letter and a unique color. Each line represents a sequence of length 64. Note that our top matches are more sensitive to displacement than matches from edit distance. **Right**: The performance of our embedding-based search, assuming that edit distance is ground truth. Here top-$k$ identifies how many top edit distance matches are considered positive out of 100,000 sample log windows. We have experimented with three values of $k$ and three log window sizes. Note that even though our embedding is not trained to mimic edit distance, it gives a useful approximation to edit distance.

edit distance. Moreover, euclidean distance is embarrassingly parallelizable and can be sped up on GPUs. However, edit distance algorithms are sequential and do not speed up as well on GPUs. In our evaluation experiment we use KD-tree to speed up nearest-neighbour search. To trade-off speed with accuracy, we retrieve five time the number of query points and pick the closest one fifth.

We ran our experiment on a dataset of 100,000 log windows. We tried log windows of length 64, 32, and 16. Given a query sequence, we take the top 10, 100 and 1000 closest sequences (based on edit distance) as positive examples and the rest as negative examples (in three different experiments). For each experiments, we we calculate average precision for 100 queries and report the mean Average Precision. Figure 6 illustrates our experiments.

## 5.7 DIAGNOSIS AND RESOLUTION RECOMMENDER SYSTEM

Many IT services providers, serve thousands of customers. These companies have a ticketing system where customers can raise their issues. These ticketing systems have a history of diagnosis and solutions to each issue. Given a ticket, A recommender system can search for the most similar issues in log history and present a statistic of what actions solved similar tickets.

## 5.8 LOG SYNTHESIS AND TEST GENERATION

Auto-regressive transformer models such as GPT-3 have been very successful in natural language generation (Brown et al. (2020)). Since TNs can handle complex relationships between tokens, they are suitable for log synthesis and test generation as well as language generation. Given a sequence of logs, a log language model can predict a probability distribution over the next log entry and sample from the distribution. This process can generate long and diverse log sequences. This process can be used to generate novel test cases for software testing.

## 6 CONCLUSION

We have presented a unified framework for carrying out several log diagnostic tasks using a pipeline of layered vector embeddings obtained by training the masked language task using the transformer architecture. Several unique aspects of log data, specifically time stamps, are incorporated into the transformer architecture. Armed with vector representations we have applied them to several downstream tasks including anomaly detection, log search and root cause analysis. We have applied our results on a real data set obtained from a leading telecommunications vendor. Our results strongly indicate that computer system diagnostics can be greatly enhanced using embeddings derived from from modern sequence to sequence models.

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
