# OpenReview forum: "Log representation as an interface for log processing applications"
_ICLR.cc/2021/Conference — Reject_

### Official Review · AnonReviewer4 · 2020-10-26
**Log Representation as An Interface for Log Processing Applications**

**Rating:** 3
**Confidence:** 4

**Review:**

This paper proposes to represent system logs at five levels of abstraction (including log sequence, parsed log, field embedding, log embedding, and sequence embedding). The representation at each level can be computed from the previous level. Transformer Network is utilized for time encoding. The paper also describes various log-related applications based on the proposed log representation. Some experiments were conducted to evaluate the proposed approach.

Logs are useful for understanding and diagnosing software intensive systems. It is good to see that this paper proposes a new neural representation of log data. The authors also suggested various applications of the proposed log representation.

In section 4, the authors only described the proposed time encoding technique, while other parts of the log representation (such as encoding a log entry) were not described. Also, it is not clear if the proposed time encoding technique is better than the related methods (there are many related methods for encoding/representing time).

The evaluation of the proposed approach is very weak. In section 5, the authors mentioned the Radio datasets. However, the use of Radio dataset is not described. The authors only evaluated the anomaly detection model on the HDFS log dataset, which is not enough. Also, the obtained results on HDFS were not very different from the results of the related work (DeepLog). Furthermore, no experiments were conducted for the causal analysis task. Therefore, the effectiveness and generalizability of the proposed approach are not clear.

The paper only compared with a few related methods for log-related tasks. Actually, this area has been widely studied and there are a lot more research work (some also utilized deep learning and language models). The authors could discuss and compare with them. Just a few examples:
Zhang et al., Robust log-based anomaly detection on unstable log data. In Proc. ESEC/FSE 2019, 807-817.
Zhu et al., Learning to log: Helping developers make informed logging decisions, in Proc. ICSE 2015. pp. 415–425.
P. He et al., “Characterizing the natural language descriptions in software logging statements,”  in Proc. ASE 2018, pp. 178–189.

---

> ### Author Response · Authors · 2020-11-25
> **Some comments in the review are not relevant. Some comments contradict verifiable facts. Please read the details.**
>
> 1- "In section 4, the authors only described the proposed time encoding technique, while other parts of the log representation (such as encoding a log entry) were not described."
>
> The reviewer is asking how transformers encode a log entries. Encoding of each log entry is learned by the Transformer network. This is a widely known characteristic of any end-to-end deep model including transformers and LSTM. Transformers learn and use a look up table for representations. Please refer to the paper "Attention is all you need". In the original submission we assumed our reviewers are aware of this. We add a note in the paper to clarify this.
>
> 2- "it is not clear if the proposed time encoding technique is better than the related methods (there are many related methods for encoding/representing time)."
>
> There are no related methods for log time encoding (!) so we could compare to. To the best of our knowledge, we presented the first work to encode “timestamp” in logs and there is NO other work to encode time for logs. We made a fresh search and we did not find any relevant works. We reuse the same formulation offered by the original transformer paper to encode location and we argue that this formulation is relevant to time as well. The original transformer paper found that static encoding (similar to what we are using) performs better than a learned encoding. So we didn't feel the need to explore alternative time encoding techniques. One can explore possible time encoding techniques in a separate work.
>
> 3- "he evaluation of the proposed approach is very weak.". We evaluated our model on two datasets, compared to four previous works and showed five downstream applications. OVER half of our paper is devoted to experiments! We also recently added a new experiment on BlueGene dataset and we outperformed a recent work.
>
> One of the papers that the reviewer himself proposed, (the only relevant one among the three papers) evaluates only on HDFS! Yet, the reviewer believes the fact that we evaluated on HDFS and Radiostation data is not enough. Also please note that we added another experiment on BlueGene data where we beat previous works.
>
> 4- ”the obtained results on HDFS were not very different from the results of the related work (DeepLog).”
>
> We outperformed DeepLog on HDFS dataset by 0.978 to 0.96. We nearly “HALVED” the error and we think the community agrees that halving error on a widely studied benchmark is considered significant.
>
> 5- “no experiments were conducted for the causal analysis task.”
>
> Section 5.5 is devoted to our experiments on causal analysis and it spans about 20% of the experiments section. We did not perform "quantitative comparison" and we devoted one paragraph to explain why. We explained that NO ground-truth and NO benchmark exists for causal analysis on logs. Therefore, there is no way anybody can perform quantitative evaluation. We established the superiority of our model on other experiments and resorted to qualitative experiments on causal analysis. Our aim was to show that causal analysis is a possible downstream applications.
>
> 6- We cited the paper “Robust log-based anomaly detection on unstable log data.” investigates anomaly detection on synthetic unstable log data. This papers uses LSTM for anomaly detection. Our model outperforms this work on the standard HDFS dataset. The 99% figure they report is on a synthetic dataset that is not relevant to us.
>
> 7- You asked to compare to the two following two papers. We carefully studied these papers and to the best of our judgement, we didn't find them relevant:
> - Zhu et al., Learning to log: Helping developers make informed logging decisions, in Proc. ICSE 2015. pp. 415–425.
> - P. He et al., “Characterizing the natural language descriptions in software logging statements,” in Proc. ASE 2018, pp. 178–189.
> These two papers investigate logging practices by software developers. There is no machine learning contribution in these two papers. The problems that these two papers investigate are not relevant to our field.
>
> Given the above rebuttals, we ask you to reconsider your rating.

---

### Official Review · AnonReviewer1 · 2020-10-27
**The paper addresses an interesting application, but lacks novelty in the approach**

**Rating:** 5
**Confidence:** 3

**Review:**

The paper deals with log data representation and analysis. Logs are important to understand the status of a system and do various root cause analysis. This paper proposes a transformer based approach to obtain vector representation of log data, in various levels such as for key-value pair with a log entry, a log entry within a sequence of logs and various blocks of log sequences. The usefulness of log embeddings are shown on multiple log downstream tasks.

The topic of the paper is very interesting and relatively less studied in machine learning domain, though important for various applications. However, it has several drawbacks, as follows.

1. The technical contribution of the paper is very limited. The transformer based approach is not novel. The specific format of time encoding in Section 4.1 is not well motivated.

2. In Section 3, is it reasonable to consider a log entry as a sequence of characters, instead of sequence of words? Most of the log entries consist of a set of key words (presented in a readable format) and some arguments (can be numeric).

3. One key difficulty in handling log data is the presence of large number of arguments (or parameters) within each log entry. Thus, it becomes difficult to mine logs to different templates. The exact parameter values can be very different in different logs. So, applying text modeling approaches directly on log data may not be the best approach.

4. In Section 3, What is the granularity used for sequence embedding in logs? Is it for a whole log file, or a block of logs? How to determine the appropriate granularity for some downstream application?

5. Fir unsupervised anomaly detection, it is not clear why anomalous points would fall into a small cluster. Rather, they would probably be distributed over multiple clusters, but still will be far away from the respective cluster center. The paper could have also used some standard anomaly detection algorithm such as Isolation Forest once the vector embeddings are generated.

6. For supervised anomaly detection, what is the % of anomalous logs used in the training?

7. Being an application-oriented paper, more importance should be given on hyperparameter setup and tuning for all the experiments. The lack of availability of source code also makes the reproducibility of the results difficult.

---

> ### Author Response · Authors · 2020-11-25
> **Three questions and four comments. Questions were answered and comments were addressed.**
>
> Thank you for your comments and questions.
>
> 1- novelty: We agree with you where “The topic of the paper is very interesting and relatively less studied in machine learning domain, though important for various applications.” This is why we realized that the community needs a paper to make a bridge between standard machine learning techniques and system logs. The novelty and the contribution of our paper is in making this connection. We found no tool better than transformer networks for processing logs. We think this paper helps people who work with logs make better use of machine learning techniques for their own applications.
>
> As a review of the novelties of our work please note that:
> - This is the first work to use transformer networks to embed sequences of logs. Prior work including Logsy and DeepLog either don't use transformers, or they only process single log entries.
> - We are the first work to encode log timestamps to learn log representation and we also show empirically for the first time that timestamps help improve embeddings. We also give a clear picture how to encode time for the first time.
> - Our paper is the first paper to review several downstream applications using a unified log embedding scheme which is important.
> - We proposed these levels of abstraction (to simplify thought process and engineering process) for the first time.
>
> 1-b- "The specific format of time encoding in Section 4.1 is not well motivated." We added a paragraph to motivate the specific format of time encoding.
>
> 2- “Is it reasonable to consider a log entry as a sequence of characters?” Our hierarchical representation provides the flexibility to do precisely that if it is warranted. In fact we have separately tried using a Fasttext like embedding (at character level) to create a clustering-based tokenizer. Furthermore, past experiments in NLP show that currently word level granularity often gives better results in practice. If in the future character level granularity outperforms word level, our methodology is compatible with that too.
>
> 3- "How to handling large number of arguments in logs":
> - At the parsing level: Our second level of abstraction is "parsed logs". The role of the log parser is exactly to extract arguments and handle grammar complexities. Log parsers parse structures as complex as JSON files and computer programs. We build our pipeline on top of log parsers so they can handle these parsing complexities.
> - At the encoding level: We don't need to "mine logs to different templates". Please note that our third level of abstraction is "field embedding". A complex log entry could have many fields that are parsed as (field_name, value) pairs. We pointed out in Section 3-3 that DeepSet combines any number of fields.
>
> 4- "In Section 3, What is the granularity used for sequence embedding in logs?" A sequence could be a whole log file, a block of logs or a sliding window. Our proposed method can handle sequence of log irrespective of what process generated those sequences. We discussed different configurations in the experiments section.
> - "How to determine the appropriate granularity?" The appropriate granularity is dictated by the application. In some applications (like HDFS) we need to identify anomalous “blocks”, so the granularity is at the block level. In search, we are interested in sliding-windows. In log-entry level anomaly detection we need log-entry level granularity.
>
> 5- “Fir unsupervised anomaly detection, it is not clear why anomalous points would fall into a small cluster.” We did not argue that anomalous points would fall into a small cluster. Rather, we argued that because anomalies are rare, anomalous points are scattered around. So anomalies form several small clusters because they are naturally scattered and cannot fall into a single cluster. One can use any clustering technique such as Isolation Forest and it may outperform k-means.
>
> Please note that our focus is not the choice of downstream clustering technique. Our focus is that we should represent logs in vector embeddings.
>
> 6- “For supervised anomaly detection, what is the % of anomalous logs used in the training?”, As noted in the paper, in the HDFS benchmark anomalies are attributed to “blocks” that contain multiple logs. In this benchmark 2.9% of blocks are anomalous. These details are the standard characteristics of HDFS benchmark and we used the same standards.
>
> 7- “Being an application-oriented paper, more importance should be given on hyperparameter setup and tuning for all the experiments.” Our paper is really a “methodology paper” rather than an “application paper”. Our goal is not to target a specific technical application, but rather we offer a methodology to learn, process and use log representations for machine learning applications. Since our focus is on methodology, we did not discuss hyperparameter details because they differ application to application.
>
> Thanks

---

### Official Review · AnonReviewer3 · 2020-10-28
**Review for Log Representation as an Interface for Log Processing Applications**

**Rating:** 4
**Confidence:** 3

**Review:**

**Summary**
Logs are widely used in computer systems to record their events. The recorded logs can be applied to a wide variety of diagnostic applications such as anomaly detection, root cause analysis, and causal analysis. This paper proposes levels of abstraction for log representation. There are in total 5 levels of abstraction, which are log sequences, parsed logs, field embeddings, log embeddings, and sequence embeddings. Each of the aforementioned levels can be derived from its proceeding levels. The paper uses the transformer model to generate a log representation. Moreover, the authors propose a time encoding method and add the encoding time to the transformer to improve the representation ability of the proposed log representation method. In the evaluation section, the authors apply the generated log representation to various downstream applications to test the effectiveness of the proposed method.

**Strengths**:
1. The idea of using different levels of abstraction to generate log embedding is interesting.
2. The paper conducts a thorough experiment over various downstream tasks based on the learned log representation.

**Weaknesses**:
1. The motivation for adding time encoding to the transformer is not clear. It would be better if the intuition behind this variation could be discussed rather than simply using the result of the ablation study.
2. It seems that the paper just applies the Transformer model to the log representation task and there is no comparison between the proposed method and other log representation methods in the evaluation part. In the related work section, the authors mentioned a recently proposed log embedding work (Logsy). It would be better if there is a comparison between the two different methods in the evaluation section.
3. There is little discussion about the results of the experiments. It would be better if some discussion of why the use of the proposed representation can or cannot improve the performance could be mentioned at the end of each experiment.

**Minor Weaknesses**:
The presentation of the paper could be improved.
1. The figures for the experiment results are too far away from the section that describes and discusses the experiment.
2. The “oh” in the caption of figure 3 should be “on”.

---

> ### Author Response · Authors · 2020-11-25
> **We addressed all of your comments.**
>
> Thanks for your helpful comments. We addressed all of your comments.
>
> 1- Motivation for time encoding: We had a quick argument in Section 2 regarding why log timestamps are helpful. We didn't elaborate more because we thought it is obvious for the readers that time information is helpful. However, we agree that further elaboration on why time is useful, is helpful. Therefore, we added two paragraphs in the beginning of Section 4 to elaborate on why time encoding helps. Here are some intuitions on why time is helpful:
>
> - An unexpected delay can signal anomaly or some underlying issue. Delays can only be seen using timestamps.
> - Event logs are often intermittent. Practical systems often record several logs within a few seconds and are silent for a few more seconds. Without time, our model has no idea how logs should be grouped temporally. With time, the attention modules in the transformer network can implicitly relate event logs that are relevant.
> - Concurrent threads/states: sometimes multiple threads write to a single log file, or multiple concurrent "states" get reflected in a single thread. Time helps the model disambiguate these threads/states.
>
> We believe there are more complex temporal patterns that humans don't understand. we leave them to be learned by the attention capabilities of the transformer model.
>
> 2- Logsy: Even though the scope of Logsy is different from our work, since you requested, we compared our model to Logsy and we outperformed it. We reflected the results in the paper.
>
> But why the scope of Logsy is different:
> - Logsy only works on "single log entry" level of granularity (our abstraction level 4). Unlike our work, Logsy does not handle sequences of logs. Therefore, Logsy cannot work on our benchmarks such as HDFS or Radio Station where what matters is “the sequence of logs”, rather than “individual logs”. Therefore Logsy does not work on the benchmarks that our work and DeepLog are evaluated on. This is why Logsy didn't report on HDFS benchmark. Logsy uses Transformer Network to process single log entries, while we use transformer Networks on sequences of logs as well. So the scope of the two works is different. However, other works such as DeepLog, PCA, IM, and N-gram are applicable to our benchmarks and we have compared to them.
> - There are a wide range of settings for anomaly detection. Logsy is a “technical paper” focusing on a specific setting for supervised anomaly detection. In contrast, our focus is not on a specific setting of anomaly detection. We are promoting a general “methodology to process logs” and we address several different applications, including but not limited to more general supervised and unsupervised anomaly detection. So again, the problems that the two works are trying to solve are different.
> - As we pointed out in Section 1.1 of our paper, the idea behind Logsy is to use logs from auxiliary datasets to improve the results on another dataset. Even though our work can be used as tool for such specific problem setting, this specific work has not been the focus of our work.
> - Even though logsy was uploaded to arxiv.org five weeks before the ICLR deadline, we cited and discussed it in our original submission. Google scholar reports that only one other paper (a survey paper) has cited Logsy yet but we cited it because they also use transformer networks.
> - After all, we beat Logsy on its own dataset of choice.
>
> 3- Discussion. As you suggested, we added discussion for each experiment. We also added an extended discussion in the supplementary material (due to space limitation).
>
> 4- We fixed the typos that you brought up.
>
> We think we addressed the three points you brought up. We elaborated on the motivation for time, we compared to Logsy and explained why its scope is different, and we added discussion on individual applications as you requested. Given that we addressed all the three weaknesses you brought up and there are no more weaknesses left, we ask you to please reconsider your rating.
>
> Thanks

---

### Official Review · AnonReviewer2 · 2020-10-31
**Log representation as an interface for log processing applications**

**Rating:** 7
**Confidence:** 4

**Review:**

This paper proposes a multi-level abstraction for representing logs that appear in a wide range of application domains and a Transformer-based model for embedding those representations in a vector space. The authors show that a variety of log processing tasks (anomaly detection, predictive analysis, search, etc.) can be implemented on top of this foundation along with empirical results for some of them based on two real-world log datasets.

Strong points:
- The paper is well written. It nicely educates the reader about the prior state of the art, motivates how the contributions of the work fit in, and then presents those contributions in a systematic manner with empirical evidence where applicable.
- Log processing is a widespread application with increasingly large and diverse datasets and need for a range of automated data analysis tasks over them. The multi-level abstraction to provide a common representation for this domain as a whole is a neat idea that can help organize existing knowledge into a common framework as well as facilitating future innovation based on that framework.
- The paper shows how transformer networks can be extended with the notion of time, which can be useful for time-oriented data in general, including logs.
- The proposed models are implemented in practice as part of a transformer library in PyTorch and are applied to a variety of log processing tasks with good results.

Weak points:
- The levels of abstraction could have been connected more clearly with the transformer model and the log processing applications in the experimental part. The mapping between these parts is not as easy to understand as it could have been based on the current writing.
- It would be good to add a figure to Section 3 to illustrate the various abstraction levels and how they are related (e.g., in the form of a pipeline together with an example log entry representation for each).

Additional comments:
- Releasing the telecommunications dataset would be a good contribution to the research community.
- Section 4.1, "Given log sequence <(l_1, t_1), ..": Please make sure to use consistent terminology. "log sequence" is defined slightly differently in Section 3.
- Figure 4: In the caption, it looks like "Middle" and "Right" should be in the opposite order. Also, the colors in the third graph are not easily readable.
- Typos:
learnt -> learned
anomaies -> anomalies
useful applications -> useful (for?) applications
oh HDFS dataset -> on HDFS dataset
pytorch -> PyTorch
figure x -> Figure x

---

> ### Author Response · Authors · 2020-11-25
> **We think this review is thoughtful and careful. Several valid issues brought up by this review and we addressed them.**
>
> Thank you for your thoughtful and careful comments. We think it is important that you noted that the goal of this paper is to promote a log analysis methodology and pipeline rather than to compete on the accuracy of a specific application.
>
> Currently the log processing community does not use standardized techniques to solve their problems. We believe the community needs a thought process to make a bridge between raw logs and machine learning applications. No standardized process is proposed before this work. By establishing a processing pipeline and a multi-layered "representation" interface, we help log processing community more easily and effectively use machine learning tools to solve their problems.
>
> Your comments:
> - We agree that we needed more connection between the levels of abstraction. As you suggested, we elaborated on the relationships between the levels of abstraction.
> - Regarding the release of the dataset: YES, we will release the communication dataset. Parts of this dataset are going to be anonymized.
> - We fixed all of the typos you pointed out.
>
> Thanks

---

### Decision · Program_Chairs · 2021-01-07
**Final Decision**

**Decision:**

Reject

**Comment:**

Most of the reviewers had concerns on the model being considered and there are additional concerns in the paper's discussion on the experimental results.